Evolutionary response to the Qinghai-Tibetan Plateau uplift: phylogeny and biogeography of Ammopiptanthus and tribe Thermopsideae (Fabaceae)

Shi Wei 1 2
Liu Pei-Liang 3
Duan Lei 4
Pan Bo-Rong brpan@ms.xjb.ac.cn 1 2
Su Zhi-Hao 1
1 Key Laboratory of Biogeography and Bioresource in Arid Land, Institute of Ecology and Geography in Xinjiang, The Chinese Academy of Sciences , Urumqi , Xinjiang , China
2 Turpan Eremophytes Botanic Garden, The Chinese Academy of Sciences , Turpan , Xinjiang , China
3 College of Life Sciences, Northwest University , Xi’an , Shaanxi , China
4 Key Laboratory of Plant Resources Conservation and Sustainable Utilization, South China Botanical Garden, Chinese Academy of Sciences , Guangzhou , Guangdong , China
Wink Michael
Electronic publication date: 2017 Jul 31
Publication date: 2017
Volume: 5
Electronic Location ID: e3607
Received 2017 Jan 25; Accepted 2017 Jul 2
Copyright: ©2017 Shi et al.
Copyright year: 2017
Copyright holder: Shi et al.
License: This is an open access article distributed under the terms of the Creative Commons Attribution License, which permits unrestricted use, distribution, reproduction and adaptation in any medium and for any purpose provided that it is properly attributed. For attribution, the original author(s), title, publication source (PeerJ) and either DOI or URL of the article must be cited.
License URL: https://creativecommons.org/licenses/by/4.0/

Keywords: Phylogeny, Biogeography, Qinghai-Tibetan Plateau uplift, Evolutionary divergence, Salweenia, Thermopsideae, Ammopiptanthus

Funding: National Natural Science Foundation of China 41271070 Special Service Project of Chinese Academy of Sciences TSS-2015-014-FW-4-1 This work was supported by the National Natural Science Foundation of China (No. 41271070) and the grants from the Special Service Project of Chinese Academy of Sciences (No. TSS-2015-014-FW-4-1). The funders had no role in study design, data collection and analysis, decision to publish, or preparation of the manuscript.

==============================
Previous works resolved diverse phylogenetic positions for genera of the Fabaceae tribe Thermopsideae, without a thoroughly biogeography study. Based on sequence data from nuclear ITS and four cpDNA regions (matK, rbcL, trnH-psbA, trnL-trnF) mainly sourced from GenBank, the phylogeny of tribe Thermopsideae was inferred. Our analyses support the genera of Thermopsideae, with the exclusion of Pickeringia, being merged into a monophyletic Sophoreae. Genera of Sophoreae were assigned into the Thermopsoid clade and Sophoroid clade. Monophyly of Anagyris, Baptisia and Piptanthus were supported in the Thermopsoid clade. However, the genera Thermopsis and Sophora were resolved to be polyphyly, which require comprehensive taxonomic revisions. Interestingly, Ammopiptanthus, consisting of A. mongolicus and A. nanus, nested within the Sophoroid clade, with Salweenia as its sister. Ammopiptanthus and Salweenia have a disjunct distribution in the deserts of northwestern China and the Hengduan Mountains, respectively. Divergence age was estimated based on the ITS phylogenetic analysis. Emergence of the common ancestor of Ammopiptanthus and Salweenia, divergence between these two genera and the split of Ammopiptanthus species occurred at approximately 26.96 Ma, 4.74 Ma and 2.04 Ma, respectively, which may be in response to the second, third and fourth main uplifts of the Qinghai-Tibetan Plateau, respectively.

Introduction

Thermopsideae (Yakovlev, 1972) is a small tribe in Fabaceae, comprising six genera, Ammopiptanthus S.H. Cheng, Anagyris L., Baptisia Vent., Pickeringia Nutt. ex Torr. & A. Gray, Piptanthus Sweet and Thermopsis R.Br. ex W.T. Aiton, with a total of ca. 45 species. Thermopsideae ranges from the Mediterranean Basin, central and northeastern Asia to temperate North America (Lock, 2005; Turner, 1981; Wang, 2001). Early phylogenetic works supported that the genera composing Thermopsideae, except for Pickeringia, were nested in the “core Genistoids” group, which always contains quinolizidine alkaloids (Crisp, Gilmore & Van Wyk, 2000; Wojciechowski, Lavin & Sanderson, 2004). A subsequent study conducted by Wang et al. (2006) resolved two clades in this tribe: the genus Ammopiptanthus clade and the “core genera” clade, consisting of Anagyris, Baptisia, Piptanthus and Thermopsis. However, Thermopsideae was not monophyletic, because Sophora nested within this tribe. Based on the plastid marker matK, recent analyses conducted by Cardoso et al. (2012a) and Cardoso et al. (2013) treated the five genera of Thermopsideae, Ammopiptanthus, Anagyris, Baptisia, Piptanthus and Thermopsis, into a narrowly defined tribe Sophoreae. However, Zhang et al. (2015a) accepted the tribe Thermopsideae and their two phylogenetic trees showed different positions of Sophora. The monophyly and genera included in the tribe Thermopsideae are thus controversial and the relationship between Thermopsideae and Sophora remains unclear. Within Thermopsideae, Anagyris (Ortega-Olivencia, 2009), Baptisia (Larisey, 1940a; Turner, 2006), Pickeringia (Wojciechowski, 2013), Piptanthus (Turner, 1980; Wei, 1998; Wei & Lock, 2010) and Thermopsis (Chen, Mendenhall & Turner, 1994; Czefranova, 1970; Larisey, 1940b; Peng & Yuan, 1992; Sa, 1999; Sa, Sudebilige & Chen, 2000) were studied taxonomically, phylogenetically and biogeographically.

Within Thermopsieae, Ammopiptanthus is a small genus, established by Cheng (1959) on the basis of A. mongolicus (Maxim.) Cheng. and A. nanus (M. Pop.) Cheng f., and being widely accepted (Yakovlev, 1988; Yakovlev, Sytin & Roskov 1996; Wei, 1998). But Wei & Lock (2010) unified the two species. Although some phylogenetic studies indicated a well-supported Ammopiptanthus (Cardoso et al., 2013; Wang et al., 2006), the infra- and inter-generic phylogeny of this genus needs further research. Zhang et al. (2015a) inferred a diverging time for Ammopiptanthus from the “core Genera” clade, but some closely related Sophoreae genera were not sampled (see Cardoso et al., 2013; Wang et al., 2006), which may have affected the accuracy of the dating.

The effects of geological and climatic factors play a key role in the spatiotemporal evolution of plants (Meng et al., 2017). The uplifts of the Qinghai-Tibetan Plateau (QTP) lead to a long-term climate oscillation in central and northern Asia. At 45–30 Ma, the collision of the Indian plate and the Asian plate triggered the first uplifting of the QTP, the Himalayan orogeny and the retreat of the Tethys (Harrison et al., 1992; Shi, Li & Li, 1999); the second main uplift (ca. 25 Ma) changed the planetary wind system and initiated the Asian monsoon (Chen et al., 1999; Li et al., 2001; Shi, Li & Li, 1999; Teng et al., 1997). The third (7–8 Ma; Harrison et al., 1992; Liu et al., 2001; Wang et al., 2008; Zheng & Yao, 2006) and fourth uplifting of QTP (3.6–2.5 Ma; Chen et al., 1999; Li & Fang, 1999; Li et al., 2001; Tang & Liu, 2001; Zheng & Yao, 2006) rendered the Asian interior cooler and drier, so evergreen forests vanished. The Tertiary broadleaf forest in Central Asia was taken over by drought-withstanding shrubs and herbs (Meng et al., 2015).

Two species of Ammopiptanthus disjunctively distributed in the southwestern Mongolian Plateau and the southwestern Pamir Plateau (Wei, 1998). Liu, Wang & Wang (1996) suggested that the ancestor of this genus emerged in the southern hemisphere, dispersing northwards when the Tertiary forest expanded due to the uplift of the QTP and the retreat of the Tethys. Subsequent studies postulated a southern Laurasian origin for Ammopiptanthus and regarded this genus as a relic of the Tertiary flora (Sun, 2002a; Sun & Li, 2003; Wang, 2001). Based on molecular evidence, Wang et al. (2006) and Zhang et al. (2015a) supported the relic status postulated for Ammopiptanthus, proposing that its ancestral area was in central Asia. However, the existing phylogeny-based biogeographic analyses used an inadequate sample of the tribe Sophoreae, which is closely related to Thermopsideae (Cardoso et al., 2012a; Cardoso et al., 2013; Azani et al., 2017), leading to possible inaccuracies in the bioinformatic inferences.

We herein employ existing GenBank sequences and newly generated sequences of the nuclear ITS and the plastid matK, rbcL, trnL-trnF and psbA-trnH gene regions, with an extensive sampling for Thermopsideae and Sophoreae, to (a) test the monophyly and systematic status of Thermopsideae; and (b) infer the phylogeny and biogeography of Ammopiptanthus.

Materials and Methods

Sampling scheme

Nine haplotypes of the two species of Ammopiptanthus found by Su et al. (2016) were included in the present study. Both species of Salweenia Baker f. were sampled (Yue et al., 2011). The nuclear internal transcribed spacer (ITS) sequences for Salweenia wardii Baker f. and Maackia amurensis Rupr. and the plastid psbA-trnH and trnL-trnF intergenic spacer sequences for Maackia amurensis were generated for the present study. The DNA extraction, amplification and sequencing methods followed Su et al. (2016). All other ITS, matK, rbcL, trnL-trnF and psbA-trnH sequences were obtained from GenBank. Guided by the phylogenetic analyses of Ammopiptanthus by Wang et al. (2006) and Zhang et al. (2015a), and the phylogeny of the Genistoids s.l. (Cardoso et al., 2012b; Crisp, Gilmore & Van Wyk, 2000; Pennington et al., 2001; Peters et al., 2010; Wojciechowski, 2003), we included all the available species of Thermopsideae and Sophoreae s.s. in our analyses. In total, we sampled 21 species in Thermopsis, seven species of Piptanthus, two species of Anagryris, six species of Bapstisia, 13 species of Sophora, three species in Maackia, two species in Euchresta and one species in each of the following genera: Ammodendron, Ammothamnus and Echinosophora. Some other species of the Genistoids s.l. were also selected according to previous phylogenetic frameworks (Cardoso et al., 2012b; Cardoso et al., 2013). Ormosia was set as the outgroup. The specific taxa, including their GenBank accession numbers, are shown in Table 1.

Table 1 Taxa names, sources and GenBank accession numbers of DNA sequences.

New sequences generated in this study are indicated by an asterisk (*). Missing sequences are indicated by a dash (–).

Species Pop.	GenBank accession number	Sources	
	ITS	rbcL	matK	psbA-trnH	trnL-trnF		
Ammopiptanthus nanus	KP636563	–	JQ820170	KP636577	KP636626		
Ammopiptanthus nanus A	KU178932	–	–	KU178934	KU178937	39.66°N, 74.75°E, 2290 m	
Ammopiptanthus nanus B	KU178932	–	–	KU178935	KU178937	39.49°N, 74.88°E, 2512 m	
Ammopiptanthus nanus C	KU178932	–	–	KU178934	KU178937	39.76°N, 76.39°E, 2350 m	
Ammopiptanthus mongolicus	KP636562	–	JQ820168	KP636576	KP636624		
Ammopiptanthus mongolicus D	KU178933	–	–	KU178936	KU178938	41.63°N, 103.22°E, 1010 m	
Ammopiptanthus mongolicus E	KU178933	–	–	KU178936	KU178939	40.49°N, 106.86°E, 1039 m	
Ammopiptanthus mongolicus F	KU178933	–	–	KU178936	KU178940	38.98°N, 105.87°E, 1762 m	
Ammopiptanthus mongolicus G	KU178933	–	–	KU178936	KU178941	37.99°N, 105.25°E, 1323 m	
Ammopiptanthus mongolicus H	KU178933	–	–	KU178936	KU178940	37.93°N, 105.26°E, 1355 m	
Ammopiptanthus mongolicus 270	KU178933	–	–	MF444199∗	MF444205∗	China: Turpan, Turpan Eremophytes Botanic Garden, Pan b. r. (TURP)	
Ammodendron conollyi	EF457705	–	–	–	–		
Ammodendron argenteum	–	–	AY386957	–	–		
Ammothamnus lehmannii	EF457706	–	–	–	–		
Anagyris foetida	AY091571	Z70122	KP230735	–	FJ499429		
Anagyris latifolia	FJ482248	–	–	–	FJ499419		
Anarthrophyllum desideratum	–	–	AY386923	–	–		
Anarthrophyllum rigidum	FJ839488	–	–	–	FJ839594		
Baptisia alba	AY773348	KP126860	KP126860	–	–		
Baptisia cinerea	AY773350	–	–	–	–		
Baptisia tinctoria	Z72314& Z72315	Z70120	–	–	AJ890964		
Baptisia sphaerocarpa	AY773351	–	–	–	–		
Baptisia australis	AY091572	KF613006	AY386900	–	FJ499421		
Baptisia bracteata	AY773349	KP126854	KP126854	–	–		
Bolusanthus speciosus	EF457708	U74243	AF142685	–	AF310994		
Bowdichia nitida	JX124478	–	JX124419	–	JX124432		
Cadia purpurea	KF850559	U74192	JX295932	–	AF309863		
Castanospermum australe	MF444193∗	–	MF444197∗	MF444201∗	MF444203∗	USA: Sri Lanka, kandy, Rudd v.e. 3339 (US)	
Calpurnia aurea	CAU59887	U74239	AY386951	–	AF310993		
Clathrotropis brachypetala	EF457714	–	–	–	AF309827		
Clathrotropis macrocarpa	–	–	JX295930	–	JX275957		
Crotalaria incana	JQ067262	JQ591662	GQ246141	JQ067481	KP691137		
Cyclolobium nutans	AF467041	–	AF142686	–	AF309857		
Cytisus scoparius	AF351120	KM360746	AY386902	–	KJ746350& AF352216		
Dicraeopetalum mahafaliense	EF457716	–	–	–	–		
Dicraeopetalum stipulare	–	–	GQ246142	–	AF310995		
Diplotropis purpurea	JX124507	JQ625878	JX124418	GQ428691	JX124441		
Echinosophora koreensis	–	AB127036	–	–	AB127028		
Euchresta formosana	–	AB127039	–	–	AB127031		
Euchresta japonica	–	AB127040	–	–	AB127032		
Genista monspessulana	JF338307	KM360800	AY386862	–	JF338219& JF338559		
Guianodendron praeclarum	JX124489	–	JX124403	–	JX124443		
Lupinus argenteus	AY338929	–	AY386956	–	AY618502& AF538706		
Maackia amurensis	MF444195∗	Z70137	AY386944	MF444200∗	MF444206∗	China: Jilin, Fusong, Sun s.n. (NENU)	
Maackia amurensis subsp. buergeri	–	AB127041	–	–	–		
Maackia chinensis	EF457721	–	–	–	–		
Maackia floribunda	–	AB127042	–	–	AB127034		
Maackia tashiroi	–	AB127043	–	–	AB127035		
Ormosia amazonica	EF457724	GQ981820	–	GQ982307	AF309484		
Ormosia fordiana	KP092737	KP094453	KP093527	KP095377	–		
Ormosia coccinea	–	JQ625915	GQ982055	GQ982308	–		
Ormosia costulata	–	–	JX295887	–	JX275917		
Pickeringia montana	MF444194∗	–	MF444198∗	MF444202∗	MF444204∗	Mexico: Tecate, Moran r. 13982 (US)	
Ormosia arborea	–	KF981227	JX295939	–	–		
Piptanthus laburnifolius	KP636565	–	–	KP636579	KP636630		
Piptanthus nepalensis	AF215922	Z70123	AY386924	–	–		
Piptanthus nepalensis1	FJ482250	–	–	KP636581	KP636631		
Piptanthus tomentosus	AY091570	–	–	–	–		
Piptanthus concolor	KP636564	–	–	KP636578	KP636629		
Piptanthus leiocarpus	AY091569	–	–	KP636580	–		
Piptanthus leiocarpus	KP636566	–	–	–	–		
Poecilanthe itapuana	KJ028462	AB045818	KJ028458	–	–		
Poecilanthe parviflora	KJ028463	–	KJ028459	–	AF208897		
Salweenia wardii	MF444196∗	U74251	–	JF725689	JF725659	China: Tibet, Qamdo, Chang et al. QZ-491 (WUK)	
Salweenia bouffordiana	–	–	–	JF725692	JF725662		
Sophora davidii	AY773352	Z70138	AY386958	JF725695	JF725665		
Sophora flavescens	FJ528290	Z70139	HM049520	JF725696	JF725666		
Sophora velutina	FN813569	–	–	–	AF309828		
Sophora jaubertii	Z72342& Z72343	Z70140	–	–	–		
Sophora macrocarpa	Z95563& Z95577	AY725479	JQ619975	–	–		
Sophora inhambanensis	FN813570	KM894237	KM896910	–	–		
Sophora tomentosa	HQ207666	AB127038	–	JX495463	AB127030		
Sophora tetraphylla	AJ310734	–	–	–	–		
Sophora howinsula	AY046514	–	–	–	–		
Sophora microphylla	AY056075	AY725480	JQ619976	GQ248391	–		
Sophora prostrata	AY056077	–	–	–	–		
Sophora raivavaeensis	AY056080	–	–	–	–		
Sophora toromiro	AY056079	GQ248696	GQ248201	GQ248392	–		
Sophora viciifolia	–	KP088855	KP089313	–	–		
Spartium junceum	DQ524327	KM360993	AY386901	HE966833	JF338264& JF338600		
Thermopsis inflata	AF123451	–	–	–	–		
Thermopsis inflata 1	–	–	–	KP636586	KP636638		
Thermopsis inflata 2	–	–	–	–	KP636639		
Thermopsis inflata 3	–	–	–	KP636587	KP636640		
Thermopsis smithiana	KP636573	–	–	KP636597	KP636650		
Thermopsis turkestanica	KP636574	–	–	KP636598	KP636651		
Thermopsis mongolica	KP636570	–	–	KP636594	KP636647		
Thermopsis alpina	KP636567	–	JQ669594	KP636582	KP636632		
Thermopsis alpina 1	AF123447	–	–	–	KP636633		
Thermopsis alpina 2	–	–	–	KP636583	KP636634		
Thermopsis alpina 3	–	–	–	KP636584	KP636635		
Thermopsis alpina 4	–	–	–	KP636585	KP636636		
Thermopsis lanceolata	AF123448	–	JQ669595	KP636589	KP636642		
Thermopsis lanceolata 1	–	–	–	KP636590	KP636643		
Thermopsis przewalskii	KP636571	–	–	–	KP636648		
Thermopsis schischkinii	KP636572	–	–	KP636596	KP636649		
Thermopsis yushuensis	KP636575	–	–	KP636599	KP636652		
Thermopsis barbata	KP636568	–	–	–	KP636637		
Thermopsis licentiana	KP636569	–	–	–	–		
Thermopsis licentiana 1	–	–	–	KP636591	KP636644		
Thermopsis licentiana 3	–	–	–	KP636592	KP636645		
Thermopsis licentiana 4	–	–	–	KP636593	KP636646		
Thermopsis turcica	JQ425645	KT175217	KT175216	KT175218	–		
Thermopsis chinensis	AF123443	–	–	GU396777	–		
Thermopsis macrophylla	AF123450	–	–	–	–		
Thermopsis divaricarpa	AY091575	–	–	–	–		
Thermopsis villosa	AY773355	–	–	–	AF311384		
Thermopsis rhombifolia	KP861904	JX848468	AY386866	KP861905	AY618487		
Thermopsis rhombifolia var. ovata	AF007468	–	–	–	–		
Thermopsis fabacea	AY091573	Z70121	–	–	–		
Thermopsis kaxgarica	–	–	–	KP636588	KP636641		
Thermopsis montana	AY091574	–	–	–	AF385411& AF385937		
Ulex europaeus	AY263686	KM361025	JQ669586	–	AF385427& AY264062		

Phylogenetic analyses

Multiple sequence alignments were performed using MUSCLE (Edgar, 2004) in the Geneious v.8.1.2 platform (Kearse et al., 2012) with default settings and manual adjustments. The best-fit substitution models for the ITS1, 5.8S, ITS2, matK, psbA-trnH, rbcL and trnL-trnF regions were determined separately using jModelTest v.2.1.7 (Darriba et al., 2012). Phylogenetic relationships were inferred using Bayesian inference (BI) as implemented in MrBayes v.3.2.5 (Ronquist & Huelsenbeck, 2003) and maximum likelihood (ML) analysis with RAxML v.8.2 (Stamatakis, 2014). The nuclear ITS dataset was partitioned into ITS1, 5.8S and ITS2 partitions. For the concatenated plastid dataset, data was partitioned separately for matK, psbA-trnH, rbcL and trnL-trnF. Two independent analyses for BI were conducted, with one cold and three incrementally heated Markov chain Monte Carlo (MCMC) chains run for 10,000,000 generations. Trees were sampled every 1,000 generations. All Bayesian analyses produced split frequencies of less than 0.01, showing convergence between the paired runs. The first 2,500 trees were discarded as burn-in and the remaining trees were used to construct a 50% majority-rule consensus tree and posterior probabilities (PP). For the ML analyses, a rapid bootstrap analysis was performed with a random seed, 1,000 alternative runs, and the same partition scheme as in the Bayesian analysis. The model parameters for each partition of the dataset were optimized using RAxML with the GTRCAT command. Trees were visualized in FigTree v1.4.3 (http://tree.bio.ed.ac.uk/software/figtree/). The ML bootstrap support values (BS) were labeled on the corresponding branches of the BI trees.

Estimation of divergence times

Divergence times were estimated using the ITS dataset and the BEAST v.2.4.3 package (Bouckaert et al., 2014). The ITS dataset was partitioned into the ITS1, 5.8S and ITS2 partitions, and nucleotide substitution models were unlinked across the three partitions. Models were determined using jModelTest. The log normal relaxed clock model was used, and the clock model was linked across partitions. The birth-death model was employed and was linked across partitions. Two independent MCMCs were each run for 50,000,000 generations, and samples were stored every 1,000 generations. The effective sample size (ESS) of each sampled parameter and the convergence between runs were checked using Tracer v.1.6 (http://beast.bio.ed.ac.uk/Tracer). The ESSs of all parameters exceeded 200, and the two independent runs were convergent. After removing a 25% burn-in from each run, the trees from the two runs were combined by using LogCombiner (Bouckaert et al., 2014). The maximum clade credibility tree was found and annotated using TreeAnnotator (Bouckaert et al., 2014), and only the branches with a posterior probability of greater than 0.5 were annotated. The dated tree was visualized in FigTree v.1.4.3.

Calibration points were chosen from the molecular dating analysis of Fabaceae conducted by Lavin, Herendeen & Wojciechowski (2005). In the matK phylogeny reported in Lavin, Herendeen & Wojciechowski (2005), the essential Genistoid crown clade (excluding Ormosia Jacks.) had been set to a minimum of 56 million years ago (Ma) according to fossil records (Herendeen & Dilcher, 1990; Crepet & Herendeen, 1992). This clade was equal to our ingroup clade; therefore, the crown age of our ingroup was set as an exponential distribution with a mean of 1 and an offset of 56 Ma. The Genistoid crown age had been estimated as 56.4 ± 0.2 Ma (Lavin, Herendeen & Wojciechowski, 2005); this age was used to set the age of the root of our tree as a normal distribution with a mean of 56.4 Ma and a standard deviation of 0.2 Ma. The age of the most recent common ancestor (MRCA) of Bolusanthus speciosus Harms and Spartium junceum Linn. was set as a normal distribution with a mean of 45.2 Ma and a standard deviation of 2.2 Ma. The age of the MRCA of Piptanthus nepalensis Sweet and Baptisia australis R.Br. was set as a normal distribution with a mean of 26.5 Ma and a standard deviation of 3.4 Ma, according to the ages of the equivalent nodes that were previously estimated by Lavin, Herendeen & Wojciechowski (2005).

Results

Phylogenetic analyses

Since plastid sequences putatively evolve as a single molecule, sequences of the four plastid markers (matK, rbcL, psbA-trnH and trnL-trnF) were concatenated. Phylogenetic analyses were conducted on both the nuclear and four combined plastid data sets (Figs. 1–3: Fig. 1 emphasized the position of Pickeringia; Figs. 2 and 3 intensified the sampling for Sophoreae). The models used in the Bayesian analyses were as follows: matK: GTR+G; psbA-trnH: HKY+G; rbcL: HKY+I+G; trnL-trnF: GTR+G; ITS1: GTR+G; 5.8S: K80+G; ITS2: GTR+G. The ITS and plastid tree topologies were distinct with regard to some key groups, thus we analyzed them separately.

Figure 1 Bayesian tree of the concatenated nuclear ITS (A) and the concatenated plastid data of matK, rbcL, trnL-trnF and psbA-trnH sequences (B) for Themopsideae and related genera.

Bayesian posterior probabilities and maximum likelihood bootstrap values are given above the branches.

Figure 2 Bayesian tree of the concatenated nuclear ITS data, showing Sophoreae and its allies.

Bayesian posterior probabilities and maximum likelihood bootstrap values are given above the branches.

Figure 3 Bayesian tree of the concatenated plastid data of matK, rbcL, trnL-trnF and psbA-trnH sequences, showing Sophoreae and its allies.

Bayesian posterior probabilities and maximum likelihood bootstrap values are given above the branches.

Our analysis (Fig. 1) showed that Pickeringia was distantly related to the Thermopsideae genera. According to the detailed trees (Figs. 2 and 3), all genera of this tribe, except Pickeringia, belonged to the well supported core Genistoids (PP = 1/BS = 100% and PP = 1/BS = 94% in Fig. 2 and 3, respectively). Four genera, Anagyris, Baptisia, Piptanthus and Thermopsis, clustered into the “Thermopsoid clade” (1/100% for ITS tree; 1/94% for plastid tree), within which Anagyris (1/100% & 1/99%) and Baptisia (1/100% & 0.95/95%) were shown to be monophyletic. The monophyly of Piptanthus was also strongly supported by the ITS tree (1/99%). Ammopiptanthus, appearing to be a sister group of Salweenia (1/100% in both trees), was monophyletic (1/100% & 0.99/89%). These two genera were not related to the Thermopsoid clade but nested in the Sophoroid clade (0.99/83% & 0.71/74%), which in turn formed a robustly supported group (1/96% & 1/100% for the tribe Sophoreae; see Discussion) sister to the Thermopsoid clade.

The sampled taxa from the tribes Crotalarieae, Genisteae and Podalyrieae formed a clade (the PCG clade; 0.89/80% & 0.92/79%), while Bolusanthus Harms and Dicraeopetalum Harms clustered together (the BOD clade; 1/100% in both trees). These two clades occupied a different position in relation to Sophoreae (0.92/88% & 0.99/56%).

Estimating divergence time

Phylogenetic dating was conducted based on the ITS dataset (Fig. 4). The estimated mean ages of the relevant clades and the 95% posterior density intervals (in parentheses) are as follows: 41.24 (35.2, 46.93) Ma for the Sophoreae plus PCG clade, 35.59 (28.88, 42.44) Ma for the Sophoroid plus Thermopsoid clade, 30.61 (22.91, 38.28) Ma for the Maackia plus its sister clade, 26.96 (19.36, 34.62) Ma for Node I, 4.74 (1.72, 8.77) Ma for Node II and 2.04 (0.67, 3.73) Ma for Node III.

Figure 4 Divergence times for Sophoreae/Themopsideae genera estimated by using BEAST based on the ITS dataset.

Calibration points are marked by A–D. Node labels and bars represent the estimated mean ages (in Ma) and their 95% highest posterior density intervals. Node I, II and III represent the divergence ages of 26.96 Ma, 4.74 Ma and 2.04 Ma, respectively.

Discussion

Phylogenetic position of Thermopsideae

Thermopsideae, the widely distributed legume tribe containing six genera, was proposed by Yakovlev (1972), and was accepted in most subsequent studies (Lock, 2005; Polhill, 1994; Turner, 1981; Wang, 2001; Wei, Gao & Huang, 2010; Wei, 1998; Yakovlev, 1972). Phylogenetic research has indicated that most genera of this tribe are members of the core Genistoids, which in turn belongs to the Genistoid clade in a broad sense (Cardoso et al., 2012b; Cardoso et al., 2016; Cardoso et al., 2013; Crisp, Gilmore & Van Wyk, 2000; Pennington et al., 2001; Peters et al., 2010; Wojciechowski, 2003). However, the western North American endemic genus, Pickeringia, was an outlier from the core Genistoids (Fig. 1; see also Lavin, Herendeen & Wojciechowski, 2005; Wojciechowski, 2013; Wojciechowski, Lavin & Sanderson, 2004; Azani et al., 2017). Therefore, Lock (2005) suggested that this genus should be excluded from Thermopsideae. Our results confirm this exclusion (Fig. 1). Pickeringia (x = 7) also differs from other genera of Thermopsideae (x = 8) in basic chromosome number (Chen, Zhu & Yuan, 1992; Goldblatt, 1981; Pan & Huang, 1993) and the absence of quinolizidine alkaloids (see Turner, 1981; Käss & Wink, 1994; Crisp, Gilmore & Van Wyk, 2000; Doyle et al., 2000; Wink, 2013).

With the exclusion of Pickeringia, Cardoso et al. (2012b) and Cardoso et al. (2013) proposed to merge Thermopsideae into Sophoreae sensu Cardoso, which is characterized by free stamens, to render it monophyletic. Merging Thermopsideae into Sophoreae is verified by our results (Figs. 2 and 3). A more inclusive Sophoreae sensu Cardoso can serve to avoid taxonomic over-fragmentation of the core Genistoids taxa and the establishments of new tribes based on many small clades. On the other hand, the clade comprising Bolusanthus speciosus Harms and Dicraeopetalum mahafaliense (M. Peltier) Yakovlev (the BOD clade), was included in Sophoreae by Cardoso et al. (2013), but was weakly supported. Such a relationship is not validated by our ITS tree (Fig. 2; it is also not supported by the likelihood bootstrap value of the plastid tree, see Fig. 3). The newly circumscribed Sophoreae, equal to Sophoreae sensu Cardoso but with the exclusion of the BOD clade, is further divided into the Thermopsoid clade and the Sophoroid clade (Figs. 2 and 3). Cardoso et al. (2013) elevated Ormosia from Sophoreae as a distinct tribe (Ormosieae), yet our results do not confirm the affiliation of Clathrotropis with this tribe (Figs. 2 and 3).

The core Genistoids is composed of three robust groups: Sophoreae, the BOD clade and the PCG clade. Our ITS and plastid tree topologies are incongruent with regard to these clades. Sophoreae forms a clade with the PCG clade in the ITS tree (Fig. 2), but the PCG clade is sister to the BOD clade in the plastid tree (Fig. 3). Although not all of the support values are significant (BI posterior probability > 0.95, ML bootstrap value > 70%), the current case of topological discordance is similar to Xu et al. (2012), García et al. (2014) and Duan et al. (2016), which likely implied a chloroplast capture event in the origin of Sophoreae. Nevertheless, highly supported phylogenetic trees based on multi-locus nuclear and plastid genes are required to further verify this hypothesis.

Phylogeny of the Thermopsoid clade

The Thermopsoid clade includes four genera: Anagyris, Baptisia, Piptanthus and a polyphyletic Thermopsis. The clade is divided into two well supported groups: the Eurasian group and the American group (Figs. 2 and 3).

The monophyletic Anagyris (also see Ortega-Olivencia, 2009) is endemic to the Mediterranean region, and belongs to the Eurasian group (Figs. 2 and 3). The Eurasian group also includes the Hengduan-Himalaya-distributed genus Piptanthus, whose monophyly was accepted by Wang et al. (2006) and supported by our ITS results (Fig. 2). Baptisia is restricted to North America (central, northern and southern states of the USA) and is embedded within the American Thermopsoid group. Our analyses yielded robust support for this genus, similar to Wang et al. (2006), Uysal, Ertuğrul & Bozkurt (2014) and Zhang et al. (2015a).

Previous studies (Uysal, Ertuğrul & Bozkurt, 2014; Wang et al., 2006; Zhang et al., 2015a) and the present results (Figs. 2 and 3) indicate a polyphyletic Thermopsis, with its species being assigned into both the Eurasian and the American groups. It is obvious that Thermopsis needs further taxonomic revision. It is noteworthy that three Asian species, Thermopsis fabacea (Pall.) DC., T. chinensis Benth. ex S. Moore and T. turcica Kit Tan, Vural & Küçük., cluster with the American group, making the biogeography of this genus an attractive topic for future research. In addition, our trees failed to support the generic status of the monotypic Vuralia Uysal & Ertuğrul (=Thermopsis turcica), which was proposed by Uysal, Ertuğrul & Bozkurt (2014) mainly based on some unique morphological characters such as a three-carpellate ovary and indehiscent fruit.

Ammopiptanthus within the Sophoroid clade

Within the Sophoroid clade, the monophyletic Maackia Rupr. diverges first, and the remaining taxa are divided into two highly supported groups. The first group contains a non-monophyletic Sophora (also see Cardoso et al., 2013; Kajita et al., 2001; Käss & Wink, 1997; Lee, Tokuoka & Heo, 2004; Wink & Mohamed, 2003) and some allied Sophoreae genera, i.e., Ammodendron Fisch. ex DC., Ammothamnus Bunge, Echinosophora Nakai and Euchresta Benn. Sophora is a widely distributed genus, and has been revised by various taxonomists (Bao & Vincent, 2010; Heenan, Dawson & Wagstaff, 2004; Ma, 1990; Ma, 1994; Tsoong & Ma, 1981a; Tsoong & Ma, 1981b; Vasil’chenko, 1945; Yakovlev, Sytin & Roskov 1996). The phylogeny and circumscription of the genus are long-standing puzzles that require considerable effort to solve.

The former Thermopsideae member, Ammopiptanthus, which is sister to Salweenia, constitutes another entity in the Sophoroid clade (Figs. 2 and 3). Traditionally, Ammopiptanthus contains two species: A. mongolicus and A. nanus (Cheng, 1959; Fu, 1987; Li & Yan, 2011; Wei, 1998; Yakovlev, Sytin & Roskov 1996), while Wei & Lock (2010) merged the latter species into the former. Our results (Figs. 2 and 3) confirmed the specific status of A. nanus, which is confined to southwest Xinjiang in China and eastern Kyrgyzstan, compared to a non-overlapping range of A. mongolicus in northern Inner Mongolia, northern Gansu, eastern Xinjiang, China and southern Mongolia (Fig. 5). Taxonomic separation of the two species is also supported by morphological (Cheng, 1959; Wei, 1998), anatomical (Yuan & Chen, 1993), cytological (Chen, Zhu & Yuan, 1992; Liu, Wang & Wang, 1996; Pan & Huang, 1993) and biochemical (Feng et al., 2011; Shi, Pan & Zhang, 2009; Wei et al., 2007; Wei & Shi, 1995; Yin & Zhang, 2004) evidence. Recently, Lazkov (2006) described a new species in Kyrgyzstan: Ammopiptanthus kamelinii Lazkov. The type specimen is not significantly distinct from A. nanus and the type locality overlapped with that of A. nanus, so we suspend the recognition of A. kamelinii.

Figure 5 Distribution (A) and representative plants of Ammopiptanthus (B & C) and Salweenia (D).

(A) red - Ammopiptanthus (I: distribution of A. mongolicus; II: distribution of A. nanus), green - Salweenia; (B) Ammopiptanthus mongolicus; (C) Ammopiptanthus nanus; (D) Salweenia wardii. Image credit for (D): Professor Zhao-Yang Chang.

On the other hand, Salweenia was originally established as a monotypic genus in Sophoreae and Yue et al. (2011) identified a second species of this genus based on morphological and phylogenetic evidence. Both species are endemic to the Hengduan Mountains in southwest China. Phylogenetic reconstruction based on the plastid rbcL sequence showed that Salweenia was sister to a Maackia-Sophora-Euchresta clade (Doyle et al., 1997). Its sistership with Ammopiptanthus is firstly discovered herein, which is further explicated as follow.

Biogeography of Ammopiptanthus and Salweenia

The abovementioned Ammopiptanthus-Salweenia group has a disjunct distribution. Ammopiptanthus is recorded from arid regions of northwest China, southern Mongolia and eastern Kyrgyzstan (Fig. 5A–5C). In contrast, Salweenia is endemic to the Hengduan Mountains in the eastern Qinghai-Tibetan Plateau (QTP) (Fig. 5A & 5D). Several hypotheses have been proposed for the evolutionary history of Ammopiptanthus, most of which suggest that this genus is a relic survivor of the Tertiary flora (Sun, 2002a; Sun & Li, 2003; Wang, 2001; Wang et al., 2006; Zhang et al., 2015a). Yet these studies were conducted in the now outdated context of Thermopsideae, rather than the more informative context of Sophoreae. Furthermore, few studies have highlighted the sister relationship between Ammopiptanthus and Salweenia.

A central Asian origin for Ammopiptanthus, as suggested by Wang et al. (2006) and Zhang et al. (2015a), may be valid if judged by the unique habit in the northwest desert of China: it is the only evergreen broadleaf shrub in this region, which can be regarded as a symplesiomorphy associated with Tertiary flora. Additionally, Salweenia is an evergreen shrub (Yue et al., 2011); this similar habit further supports its sister relationship status with Ammopiptanthus. Due to the monophyly of the Ammopiptanthus-Salweenia group, the ancestral range of Salweenia is probably not in Gondwana as described in Li & Ni (1982) and Yue et al. (2011). Thus, we hypothesize the evolution of this group as described below (see Fig. 4). The Himalayan orogeny and uplifting of the QTP initiated the retreat of the Tethys (ca. 45–30 Ma; Harrison et al., 1992; Shi, Li & Li, 1999; Zhang & Fang, 2016). The second major uplift of the QTP occurred at ca. 25 Ma, triggering the East Asian monsoonal climate of the Asian interior, including Central Asia, northwestern China and the Mongolian Plateau, which began to fluctuate, though evergreen forest temporarily remained (Teng et al., 1997; Chen et al., 1999; Shi, Li & Li, 1999; Li et al., 2001; Zhang & Fang, 2016). The common ancestor of Ammopiptanthus and Salweenia arose in the Tertiary evergreen forest of ancient Central Asia (the north coast of the Tethys) before 26.96 Ma (Fig. 4: Node I). During the expansion of the Central Asian evergreen forest, this common ancestor probably dispersed southwards along new land that emerged from the Tethys (as in Sun, 2002b).

The third rapid uplift of the QTP happened 7–8 Ma (Harrison et al., 1992; Liu et al., 2001; Wang et al., 2008; Zheng & Yao, 2006) and was followed by a major raising of the northwest QTP at ca. 4.5 Ma (Zheng et al., 2000), causing a cooler climate and aridification of the Asian inland. The Tertiary forest gradually gave way to psychrophytic and xerophytic shrubs and herbs (Sun, 2002a; Meng et al., 2015). This dramatic environmental change possibly led to the divergence between Ammopiptanthus and Salweenia (ca. 4.74 Ma, see Fig. 4: Node II). The former, remained in the Asian interior, kept the evergreen shrubby habit, and acquired xeric characters, such as the pubescent, coriaceous leaves, in the arid central Asian habitat; while the latter retained more traits from Tertiary flora in the less disturbed and wetter region of the Hengduan Mountains (Sun, 2002a; Sun, 2002b; Sun & Li, 2003).

The split of the two Ammopiptanthus species (2.04 Ma; see Fig. 4: Node III) is possibly a response to the last (fourth) rapid uprising of the QTP, when aridification of the Asian interior intensified and the Loess Plateau formed, which potentially served as a geological barrier and facilitated speciation (3.6-2.5 Ma; Chen et al., 1999; Li & Fang, 1999; Li et al., 2001; Tang & Liu, 2001; Zheng & Yao, 2006). This estimated age is slightly older than that proposed in Su et al. (2016), who similarly suggested that the speciation of Ammopiptanthus was caused by climate oscillation and range shifts. Ammopiptanthus nanus grows in a dryer habitat than that of A. mongolicus; the former, therefore, possesses more xeric apomorphies such as shorter habit, usually 1-foliolate leaves, conspicuous leaf venation, thicker root cortex, more complex karyotype and more vulnerable phytocommunities (Cheng, 1959; Pan & Huang, 1993; Wei, 1998; Zhang et al., 2007).

Such disjunction resulting from the QTP uplift can be found in other Fabaceae species. Examples are the infra-generic biogeography of some genera in the tribe Caraganeae (QTP-NW China/C Asia disjunction; see Zhang et al., 2010; Zhang et al., 2015b; Zhang et al., 2015c) and the inter-generic evolutionary history of Gueldenstaedtia and Tibetia (mesic E Asia-QTP disjunction; see Xie et al., 2016). Our results may provide new insight into the evolutionary pattern of an inter-generic QTP-Asian interior disjunctive distribution.

Conclusion

Thermopsideae is a widely spread tribe of Leguminosae, ranging in temperate Eurasia and North America, its phylogeny has been controversial for decades. According to our results, Pickeringia was excluded from Thermopsideae. The previous finding, that this tribe is part of an expanded Sophoreae, was confirmed herein. The re-delimited Sophoreae contained two clades: Thermopsoid and Sophoroid clade. Monophyly of Anagyris, Baptisia and Piptanthus were supported in the former clade. On the other hand, Ammopiptanthus, including A. mongolicus and A. nanus, nested within the Sophoroid clade, with Salweenia as its sister. The Ammopiptanthus-Salweenia clade displayed a disjunctive distribution in northwestern China-central Asia and Hengduan Mountains, respectively. The estimation of divergence ages showed the emergence of the common ancestor of Ammopiptanthus and Salweenia, divergence between these two genera and the split of Ammopiptanthus species are in response to the second, third and fourth main uplifts of the QTP, respectively.

Supplemental Information

Supplemental Information 1 DNA sequence alignments used in this study

The nuclear ITS and combined plastid datasets are given in fasta format.

Click here for additional data file.

We thank Dr. Ming-Zhou Sun for kindly providing samples and Prof. Zhao-Yang Chang for providing the photograph in Fig. 5D.

Additional Information and Declarations

Competing Interests

Author Contributions

Data Availability

The authors declare there are no competing interests.

Wei Shi conceived and designed the experiments, performed the experiments, contributed reagents/materials/analysis tools, wrote the paper, prepared figures and/or tables, reviewed drafts of the paper.

Pei-Liang Liu conceived and designed the experiments, performed the experiments, analyzed the data, contributed reagents/materials/analysis tools, wrote the paper, prepared figures and/or tables, reviewed drafts of the paper.

Lei Duan performed the experiments, analyzed the data, contributed reagents/materials/analysis tools, wrote the paper, prepared figures and/or tables, reviewed drafts of the paper.

Bo-Rong Pan was the project leader.

Zhi-Hao Su performed the experiments.

The following information was supplied regarding data availability:

The raw data has been supplied as Supplemental Information 1.

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
