# Peer review of "Evolutionary response to the Qinghai-Tibetan Plateau uplift: phylogeny and biogeography of Ammopiptanthus and tribe Thermopsideae (Fabaceae)"

_PeerJ, doi:10.7717/peerj.3607_

## Round 0.1 · original submission · Major Revisions

· Academic Editor

Major Revisions

Dear authors

As you can see from the comments of our reviewers your ms has several problems which need to be fixed. Be aware that we will send the revised ms to the same reviewers.

Kind regards,

Michael Wink
Academic editor

·

Basic reporting

The paper is basically sound but some limited editing by an English-speaking person would be necessary. I have already proposed many improvements (see attached pdf with track marks).

The Reference list is absolutely riddled with small errors and inconsistencies. I have marked some examples in the pdf but unfortunately have no time to proofread - this is the authors' job!

A little more background is needed (as inserted into the pdf) to avoid creating the impression that the analysis was based on new DNA data. Practically all the sequences were from GenBank, and the reader need to know this upfront.

The results mainly support previous studies, especially Cardoso (2013) and Zhang (2015). This fact should be better explained in the text, e.g. "The concept of the Qinghai-Tibetan Platea upliftment and its role in the evolutionary history of Thermopsideae was first proposed by Zhang (2015) and is here further explored." The cladogram for the Sophoreae in Cardoso et al. 2013, includes fewer species but is hardly different from the one presented in this paper.

Zhang ML, Huang JF, Sanderson SC, Yan P, Wu YH, and Pan BR. 2015. Molecular Biogeography of Tribe Thermopsideae (Leguminosae): A Madrean-Tethyan Disjunction Pattern with an African Origin of Core Genistoides. BioMed research international 2015:864804. 10.1155/2015/864804

Experimental design

The paper contains hardly any new DNA sequences, so that it is basically a new analysis of existing genbank sequences. This should be explicitly stated, otherwise the reader will be falsely led to the assumption that the sequences referred to are new.

The main knwledge gap is the position of Salweenia (and perhaps a more thorough analysis of Ammopiptanthus). However, the position of Salweenia is hardly a surprise and was anticipated by previous researchers. Practically all the other results merely confirm (or perhaps slightly refine) what others have already proposed.

Validity of the findings

The novelty is relatively low, because almost all findings merely validate what previous researchers have already found. The paper can be improved by explaining clearly what aspects in the present paper are really novel/original. As far as I can see, it is only the position of Salweenia that is really a new contribution, but this is not unexpected, given other characters (already anticipated by Cardoso et al. in the 2013 paper).

The data and analyses seem robust and sound.

More recognition (in the form of citations) can be given to the originators of the concepts which have here been elaborated upon, to make it clear that the concepts are not new.

Comments for the author

It will be important to clearly distinguish between the hypothesis and findings of Zhang et al. (2015) and also Cardoso et al. (2013) so as not to create the impression (inadvertently) that the "Upliftment Theory" is a new idea, or that the phylogeny for the Sophoreae-Thermopsideae group is new. Please explicitly state what are new data, and what findings are really new.

·

Basic reporting

Comments:

The paper by Shi et al. presents a molecular phylogenetic, biogeographic analysis of Ammopiptanthus. They found 2 distinct species which are sister to Salweenia.
The study is mainly focussed on the Central/eastern Asian biogeographic relationsships. Technically the study is adequately done. I would suggest to basically partly restructure and strengthen the text as major revisions.

The problem on the uplifts of the Qinghai-Tibetan Plateau and the relationships the Mongolian deserts should be worked out more in detail. An own paragraph in the intro describing the geological processes in more detail and the ecological consequences would be necessary. These data can be picked up in the discussion again. It would make the text more stringent. And makes the reader clear why your ms. is interesting.
Don’t only cite some other papers, but give a small summary on their thoughts.

An important point for Dating is described below.



l. 20 why basically?
l. 25-26. Describe which crown or stem node has which age. In the present form it is not so clear.
l. 37 by Wang
l. 38 unsisterly what is this?
l. 52 needs
l. 56 As for the phytog.
l. 62-69 Make clear what is the new thing of your study. Rephrase.
l. 82 so 21 out of 45 species were covered. May any of the unsampled taxa belong to the Ammopiptanthus group based on morphological characters? Discuss.
l. 84 an so on => describe in detail
l. 125 ! using a normal distr. 0f 56.4 +- 0.2 Ma makes the calibration point really narrow. However, Lavin published these data. Are there any other alternative dating results? Perhaps newer ones?
l. 201/292 rephrase: highly supported analyses?
l. 215-217 rephrase
l. 241 summarise the morph. Characters.
l. 247 discuss why Lazkov distinguished A. kamelinii from other taxa.
l. 258 conclusion of the paragraph?
l. 259-262 discuss why the habit should be sysmplesiom.
l. 273 why vicariance? Explain.
l. 285- shorter plants? Explain more comples karyotype. Explain ‘etc.’
Overall discuss possible alternatives to your interpretation to the uplift.

Experimental design

see Basic reporting

Validity of the findings

see Basic reporting

Comments for the author

see Basic reporting

---

## Round 0.2 · Minor Revisions

· Academic Editor

Minor Revisions

Please follow the remaining minor advice of the reviewer

MWink

·

Basic reporting

This is a review on a revised version of the paper by Shi et al. presenting a molecular phylogenetic, biogeographic analysis of Ammopiptanthus.

The text is now much better than in the 1st version.
I would still suggest that the author add a few points to the discussion:
1) Please discuss the dating. The calibration is very narrow ( see 1st review), so the outcomes may change when using other calibration priors.
2) The taxa may be a nice example that follows the QTP uplift. Still there are always alternative explanations. Please explain why you exclude such other explanations like long distance dispersal or alternative ancestral states.

I would add a terminal conclusive paragraph.

After implementation of these few points I would regard the ms. as acceptable.

I could not find the figs. in the pdf. I assume they were unchanged.

l. 51 positions
l. 64 delete ‘always’
l. 78 tertiary => Tertiary, basically it is split into Neo and Paleogene.
l. 307 suggest
l. 348 what is psychric? Might not be widely used.
l. 350 probably => possible might be better.

Experimental design

See Basic reporting

Validity of the findings

See Basic reporting

Comments for the author

See Basic reporting

---

## Round 0.3 · Minor Revisions

· Academic Editor

Minor Revisions

Thanks for adjusting the ms according to the recommendations of the reviewers.

I just read the revision and found a few more minor items which need your attention:

References:
Käss not Kass

Scientific names: check for typos; eg. Maackia not Maakia, Capitals for family names; no italics for author names (in scientific names)

The polyphyly of Thermopsis is important; please mention it in the abstract as well

Please check the following papers which are relevant for your work:
71. The Legume Phylogeny Working Group (LPWG): Nasim Azani,et al A new subfamily classification of the Leguminosae based on a taxonomically comprehensive phylogeny. TAXON 66: 44–77
Wink, M. (2013) Evolution of secondary metabolites in legumes (Fabaceae). South African Journal of Botany, 89, 164–175

[# Staff Note: It is PeerJ policy that any additional references suggested during the review process should only be included if the authors agree they are relevant and appropriate. #]

Kind regards

M. Wink

---

## Round 0.4 · Minor Revisions

· Academic Editor

Minor Revisions

Dear authors

Thanks for revising the MS, but there are still items to be fixed

it is Käss with an umlaut not Kass! please correct

Question of polyphyly:

You should mention in the abstract that the genera Thermopsis and Sophora show substantial polyphyly, which requires a thorough taxonomic revision.

(not Thermopsideae, as you wrote)

The citation I gave was from 2017

N Azani, M Babineau, C. D. Bailey, H Banks, A R. Barbosa, R Barbosa Pinto, J S. Boatwright, L M. Borges, G K. Brown, A Bruneau, E Candido, D Cardoso, K-F Chung, R P. Clark, A de S. Conceição,, M Crisp, P Cubas, A Delgado-Salinas, K G. Dexter, J J. Doyle, J, Duminil, A N. Egan, M de la Estrella, M J. Falcão, D A. Filatov, A Pa Fortuna-Perez, R H. Fortunato, E Gagnon, P Gasson, J Gastaldello Rando, A-M Goulart de Azevedo Tozzi, B Gunn, D Harris, E Haston, J A. Hawkins, P S. Herendeen, C E. Hughes, J R.V. Iganci, F Javadi, S A Kanu, S Kazempour-Osaloo, G C. Kite, B B. Klitgaard, F J. Kochanovski, E J.M. Koenen, L Kovar, M Lavin, M le Roux, G P. Lewis, H C. de Lima, M C López-Roberts, B Mackinder, V Hugo Maia,V Malécot, V F. Mansano, B Marazzi, S Mattapha, J T. Miller, C Mitsuyuki, T Moura, D J. Murphy, M Nageswara-Rao,B Nevado, D Neves, D I. Ojeda,R. T Pennington, D E. Prado, G Prenner, L Paganucci de Queiroz, G Ramos, F L. Ranzato Filardi, P G. Ribeiro, M de Lourdes Rico-Arce, M J. Sanderson, J Santos-Silva, W M.B. São-Mateus, M J.S.Silva, M F. Simon, Ce Sinou, C Snak, É R. de Souza, J Sprent, K, P. Steele, J E. Steier, R Steeves, C H. Stirton,S Tagane, B M. Torke, H Toyama, D Trabuco da Cruz, M Vatanparast, J J. Wieringa, M Wink, M F. Wojciechowski, T Yahara, T Yi, E Zimmerman 2017 A new subfamily classification of the Leguminosae based on a taxonomically comprehensive phylogeny. TAXON 66: 44–77

Regards
MWink

---

## Round 0.5 · accepted · Accept

· Academic Editor

Accept

Nihao

Congratulations! Thanks for the revisions. Your ms can now be accepted.

Greetings
Michael Wink
Academic editor